# Nicotinamide Mononucleotide Supplementation: Understanding Metabolic Variability and Clinical Implications

**DOI:** 10.3390/metabo14060341

**Published:** 2024-06-18

**Authors:** Candace Benjamin, Rebecca Crews

**Affiliations:** Renue By Science, Jacksonville, FL 32256, USA; candace_benjamin@outlook.com

**Keywords:** nicotinamide mononucleotide, NAD+, vitamin B3, human longevity, bioavailability

## Abstract

Recent years have seen a surge in research focused on NAD+ decline and potential interventions, and despite significant progress, new discoveries continue to highlight the complexity of NAD+ biology. Nicotinamide mononucleotide (NMN), a well-established NAD+ precursor, has garnered considerable interest due to its capacity to elevate NAD+ levels and induce promising health benefits in preclinical models. Clinical trials investigating NMN supplementation have yielded variable outcomes while shedding light on the intricacies of NMN metabolism and revealing the critical roles played by gut microbiota and specific cellular uptake pathways. Individual variability in factors such as lifestyle, health conditions, genetics, and gut microbiome composition likely contributes to the observed discrepancies in clinical trial results. Preliminary evidence suggests that NMN’s effects may be context-dependent, varying based on a person’s physiological state. Understanding these nuances is critical for definitively assessing the impact of manipulating NAD+ levels through NMN supplementation. Here, we review NMN metabolism, focusing on current knowledge, pinpointing key areas where further research is needed, and outlining future directions to advance our understanding of its potential clinical significance.

## 1. Introduction

NAD+, short for nicotinamide adenine dinucleotide, is a coenzyme found within every living cell with vital roles in various fundamental biological processes, including but not limited to, energy metabolism, DNA repair, cellular communication, and enzymatic activation. Originally, NAD+ was identified for its role in enhancing fermentation in yeast, where researchers found evidence of a coenzyme whose presence was crucial for alcoholic fermentation [1]. Though this early research predates our current understanding of NAD+, it prompted diverse biological investigations into its role in cellular redox reactions [2], electron transfer [3], glycolysis [4], the Krebs cycle [5], and fatty acid β-oxidation [6].

Beyond its metabolic functions, NAD+ acts as a coenzyme for vital proteins including sirtuins [7], polyadenosine diphosphate-ribose polymerases (PARPs) [8], and CD38 [9,10], participating in molecular deacetylation [11], ribosylation, glycohydrolase synthesis [9], and NAD+-dependent signal transduction. These actions contribute to metabolic homeostasis and cell signaling, highlighting the continuous demand for NAD+ within the body. Unsurprisingly, the mechanism of NAD+ production varies slightly across species but ultimately aims to maintain a delicate balance between production and recycling to ensure proper cellular function. Emerging research now suggests a close link between NAD+ levels, aging, and an increased risk of age-related diseases, such as neurodegenerative disorders [12], cellular senescence [13], and cardiovascular complications [14]. An analysis of human skin samples revealed an age-related accumulation of oxidative DNA damage, increased lipid peroxidation, and a significant (~70%) decline in NAD+ levels [15]. Data collected on human livers indicate a 30% loss between the ages of 45–60+, and two independent MRI studies [16,17] of the brain revealed a 10–25% decline in NAD+ levels from adolescence to old age. These findings align with observations in cell and rodent models, reinforcing the importance of understanding NAD+ metabolism in aging and prompting its progression to human clinical trials to evaluate whether the effects of aging can indeed be reduced or even reversed.

The mechanisms underlying the touted beneficial effects of NAD+ precursors are centered around their influence on cellular metabolism, stress response, and DNA repair through ensuring sufficient NAD+ activity. Administration of these precursors and subsequent increase in NAD+ enhances the activity of key enzymes in the citric acid cycle, sirtuin, and PARP activation. This translates to enhanced cellular resilience against oxidative stress, DNA damage, and other hallmarks of aging. However, further research is necessary to fully elucidate the specific mechanisms by which NAD+ precursors exert their effects in different tissues and cell types.

Increasing NAD+ levels is a logical first step to remediating the natural age-related decrease of the coenzyme, and this can be achieved in a facile manner through exogenous intake, a healthy diet, and exercise [18]. B vitamins and tryptophan from dietary intake contribute to a pool of NAD+ precursors, but the majority of NAD+ is biosynthesized from internally generated and recycled precursors. In this context, nicotinamide mononucleotide (NMN) emerges as a promising candidate for NAD+-boosting interventions. As a direct precursor to NAD+, NMN readily converts within cells, potentially mitigating the age-related NAD+ decline. This potential has generated significant interest in its role as a putative anti-aging strategy, requiring rigorous scientific exploration to fully understand the safety, efficacy, and long-term consequences of NMN supplementation and the subsequent effects of NAD+ boosting in humans.

## 2. NAD+ Biosynthesis Pathways

Three independent pathways, meticulously controlled by unique sets of enzymes and precursors, ensure sufficient NAD+ levels within its specific compartments: cytoplasm, mitochondria, and nucleus [19] (Figure 1). These pathways maintain an intricate balance of NAD+ synthesis, usage, recycling, and regeneration.

Serving as the predominant mechanism for maintaining cellular NAD+ homeostasis, the salvage pathway efficiently recycles NAD+ precursors, primarily utilizing nicotinamide (NAM) generated by the enzymatic breakdown of NAD+ in various cellular processes [20]. Dietary sources of nicotinamide riboside (NR) and NMN can also contribute to this pathway. Within this pathway, NMN serves as the key intermediate. To initiate NAD+ production, NMN is reportedly transported into cells via the recently discovered Slc12a8 transporter [21]. Alternatively, NR enters cells through equilibrative nucleoside transporters (ENTs) and is converted into NMN by NR kinases (NRKs) in one step [22]. NMN is then efficiently converted to NAD+ by the enzyme NMN adenyltransferase (NMNAT). The cycle continues with NAD+-consuming enzymes, such as CD38 and sirtuins, which release NAM in the reaction. NAM phorphorylribotransferase (NAMPT), the rate-limiting enzyme in this pathway, then catalyzes the conversion of NAM back to NMN, perpetuating the salvage loop. This process enables swift NAD+ production and continuous replenishment of precursors, independent of external resources. 

The two remaining NAD+ biosynthetic pathways rely on dietary precursors, which can be advantageous when salvageable precursors are limited. The Preiss-Handler mechanism begins with nicotinic acid (NA)—a form of vitamin B3 or niacin found in fish, poultry, nuts, grains, and vitamin supplements. In three steps, NA is transformed into NA mononucleotide (NAMN), followed by NA adenine dinucleotide (NAAD), and finally to NAD+ by nicotinate phosphoribosyl transferase (NAPRT), NMNAT, and glutamine-dependent NAD+ synthetase (NADS) respectively [23,24]. Similar to the Preiss-Handler pathway, the *de novo* pathway relies on dietary tryptophan [25] which enters cells via specific amino acid transporters, LAT-1 and hPAT4 [26,27]. The enzymes indoleamine 2,3-dioxygenase (IDO) or tryptophan 2,3-dioxygenase (TDO) then convert tryptophan to N-Formylkynurenine. A series of five subsequent enzymatic reactions lead to the formation of quinolinic acid (QA). Finally, the enzyme QPRT catalyzes the conversion of QA into NAMN, which can then feed into the Preiss-Handler for NAD+ production. The liver is the primary site for NAD+ synthesis from tryptophan, as most cells lack the enzymes required for the *de novo* pathway [28]. Consequently, the majority of cellular tryptophan is metabolized to NAM within the liver, released into circulation, and subsequently taken up by peripheral cells for conversion to NAD+ via the salvage pathway [29].

Boosting cellular NAD+ levels can be achieved through supplementation with NAD+ precursors that directly participate in NAD+ synthesis, without the need for lengthy conversion pathways. As a direct precursor, NR is one such candidate that is readily converted to NMN by NRKs, quickly integrated into the salvage pathway, and is well tolerated in moderate doses [30,31]. Preclinical studies show promise of its efficacy. NR is reported to have a short elimination half-life and the effects of long-term administration are still under investigation [32]. One step closer to NAD+ in the salvage pathway is NMN, which has been shown to increase NAD+ levels in as little as 30 min and is safe during one year of chronic dosing in mice [33]. NMN also appears to be more stable in water, and when taken orally, is rapidly absorbed, and metabolized, with many studies showing increased NAD+ biosynthesis and the improvement of several age-related ailments [34]. In murine models, NMN is studied extensively and has displayed several benefits such as improved mitochondrial metabolism, insulin secretion, and ischemia, to name a few [35,36]. While translating pre-clinical findings to humans is always a challenge, successfully applying data from animal models could pave the way for groundbreaking solutions for age-related conditions. This data is crucial to establish NMN as a viable NAD+ boosting precursor and determine to what extent previous clinical data is applicable.

## 3. Impact of Metabolism on Supplementation

Supplementation with NAD+ precursors offer a promising therapeutic approach to address declining NAD+ levels. However, a major challenge lies in the limited bioavailability of orally administered NAD+ precursors due to extensive metabolism in the gastrointestinal (GI) tract and liver.

### 3.1. Gastrointestinal Tract

The impact of gut bacteria on NAD+ precursor metabolism and the complexities of cellular uptake mechanisms have been extensively researched in recent years. Significant progress has been made, but interesting questions remain and warrant further investigation.

#### 3.1.1. Interaction with Gut Microbiome

NAD+ precursors can be categorized into two chemical groups: amidated (NR, NMN, NAM) and deamidated (NA, NAMN, NAAD). Traditionally, these groups were thought to follow distinct biosynthetic routes. Amidated precursors were known to follow the salvage pathway (NR → NMN → NAD+), while deamidated precursors fueled the Preiss-Handler pathway (NA → NAMN → NAAD → NAD+). However, this paradigm shifted with the observation of increased NAAD (a deamidated intermediate) in blood cells after NR (an amidated precursor) supplementation [37]. This unexpected finding hinted at an interplay between the pathways, potentially involving previously unrecognized enzymatic activities in mammals for NAD+ production. Prior research identified a gut bacterial enzyme, PncC, capable of converting NMN into NAMN (a deamidated precursor) for NAD+ synthesis via the Preiss-Handler pathway [38]. This suggests a potential role for gut bacteria in NAD+ metabolism and could explain the rise in NAAD observed after NR supplementation.

Subsequent studies confirmed the crucial role of the gut microbiome in bridging the amidated and deamidated NAD+ pathways. In mice, orally administered NAM was deamidated by the microbial nicotinamidase, PncA, generating NA, nicotinic acid riboside (NAR), and NAAD [39]. This microbial deamidation process was important for NAD+ synthesis in the colon, liver, and kidney. Similarly, another study showed that gut bacteria facilitated the deamidation of orally administered NMN, yielding the metabolites NAR and NAMN (in the gut and liver), and NAAD (in the liver) for incorporation into NAD+ via the *de novo* pathway [40]. In germ-free mice, NMN escaped microbial deamidation and instead fueled the salvage pathway, leading to elevated levels of amidated metabolites, NR and NMN. 

Surprisingly, antibiotic treatment doubled gut NAD+ metabolite levels (including NMN, NR, NAD+, and NAM) in mice, even without NMN supplementation. This finding suggests competition by gut microbiota for both dietary and endogenous NAD+ sources. Further highlighting the complexity of NAD+ metabolism, Yaku et al. revealed a two-step process in NR utilization. Initially, direct NR uptake occurred in the small intestine for up to an hour, driving NAD+ synthesis through the classic salvage pathway even in the presence of gut bacteria [41]. Following this, the enzyme bone marrow stromal cell antigen 1 (BST1) hydrolyzed NR to NAM, which was then metabolized to NA by gut microbiota, fueling the Preiss-Handler pathway and becoming the major driver of NAD+ production. They also found that BST1 transformed NR into NAR through a base-exchange reaction using NA and NAM, providing another connection between amidated and deamidated precursors. 

A key finding of these studies is that orally administered NR, NMN, and NAM predominantly undergo degradation in a process dependent on gut microbiota, resulting in the formation of NAM or NA and downstream deamidated metabolites. Only a minor portion of NAD+ precursors delivered orally integrate into tissues without significant alteration (Figure 2).

#### 3.1.2. Uptake Mechanisms

Two primary pathways are currently recognized for NMN entry into enterocytes: an indirect route involving conversion to NR, and a direct route mediated by a specific NMN transporter. However, the relative importance and specific contributions of each pathway have not been determined. Initially, the prevailing model suggested NMN relies solely on an NR-mediated pathway. Here, the extracellular enzyme CD73 converts NMN to NR, which is then imported by equilibrative nucleoside transporters (ENTs) and phosphorylated back into NMN by NRK1/2 enzymes within the cell. Studies support this pathway, demonstrating the requirement of NRKs for NMN to boost NAD+ levels in muscle and liver cells [22,42]. However, the slow processing time of this pathway, exceeding several hours in some studies, cannot account for rapid NMN absorption within the gut (2–3 min) and tissue uptake (10–30 min) observed in mice [33]. Additionally, robust NMN uptake in cells despite inhibition of CD73/ENT or NRK1 further suggests an alternative route exists [22,43]. The discovery of Slc12a8, a highly specific NMN transporter abundantly expressed in the gut, pancreas, liver, and white adipose tissues, provides a compelling mechanism for rapid NMN absorption and distribution. Studies demonstrate that Slc12a8 deletion in these organs significantly reduces NMN uptake and NAD+ levels, supporting its role in direct NMN transport [21,44].

A published response to the initial study challenged the validity of the Slc12a8 findings, indicating that inappropriate methods were used in the study. Agreement on accurate NMN quantification methods continues to be a challenge in NAD+ metabolism research today, due to the lack of standardized protocols. Grozio et al. provided evidence supporting the chosen analytical technique in a subsequent publication [21,45]. Data interpretation was also contested, as the researchers adapted the protocols to address the inherent instability of the NMN molecule. The critique further argues that previous data showed sufficient evidence to conclude NMN was transported into the cell via conversion to NR, not via the Slc12a8 transporter. However, a key difference in the time of sampling (minutes vs. hours) indicated both routes of uptake are feasible. Tracer studies on NMN metabolism also yield inconsistent results, as some suggest significant NMN conversion to NR in the gut, with limited direct absorption [40,46]. In contrast, a study detected NMN in the intestine within 10 min of oral intake, supporting the role of Slc12a8 in direct transport [33]. Other studies have detected NMN in the intestine within 10 min of oral intake and minimal NMN-to-NR conversion in the gut when measured within minutes [33,44]. 

The relative contributions of NRK1/2 and Slc12a8 to NMN uptake mechanisms likely vary depending on time, cell type, tissue, and physiological conditions. This dynamic interplay was illustrated in septic mice, where NRK1/2 enzyme levels decreased significantly, while Slc12a8 expression remained stable [47]. Notably, NMN supplementation still effectively increased NAD+ levels despite NRK1/2 pathway suppression, underscoring the critical role of alternative, NR-independent uptake mechanisms such as Slc12a8. Tissue-specific expression patterns further contribute to this complexity. Organs with inherently low NRK1 activity, such as the heart and white adipose tissue, might predominantly rely on Slc12a8-mediated uptake for NAD+ production [22,48,49]. This could be due to differences in metabolic demands, regulatory factors, or unique cellular processes within these tissues, warranting further investigation.

### 3.2. Portal Delivery

After absorption in the intestine, NAD+ precursors flow to the liver via the portal vein. Analysis of portal blood at three hours after oral NAM administration in mice revealed the presence of NAMN, NAR, NA, NMN, and NAM. While the major deamidated NMN metabolites, NAMN and NA, were detected, their concentrations were significantly lower (100–400-fold) compared to NAM, the predominant circulating precursor [29,39]. Four hours after NR gavage, elevated levels of NA and NAR appeared in portal blood, indicating bacterial contribution to NR metabolism. These metabolites were absent in germ-free mice, highlighting the essential role of gut microbiota. Interestingly, germ-free mice also exhibited a diminished increase in circulating NAM, further emphasizing the microbiota’s influence on NAD+ precursor processing [39].

### 3.3. Hepatic First-Pass Metabolism

NMN undergoes extensive first-pass metabolism in the liver when administered orally. Studies across various dosages (50 mg/kg to 500 mg/kg) consistently show that nearly all ingested NMN is converted into NAM within the liver [29,40,46]. This minimizes the amount of intact NMN available for direct NAD+ synthesis in the liver and prevents it from reaching peripheral tissues, significantly impacting overall bioavailability. 

Bypassing gut metabolism through intraperitoneal (IP) injection offers a different approach. In mice injected with radiolabeled NMN (500 mg/kg), the NMN reaches the liver first via the portal vein. While some newly generated NAD+ in the kidneys and small intestines showed incorporation of labeled NR (suggesting localized NMN conversion to NR before NAD+ synthesis), most tissues relied on NMN-derived NAM for NAD+ production. Minimal intact NMN utilization was observed in the kidney and white adipose tissue [46]. It’s important to note that this study only assessed NMN incorporation at two and four hours, potentially missing early metabolic events. For example, a separate study using IP injection of NMN (500 mg/kg) observed that mice displayed a rapid uptake, with liver NMN levels surging 15-fold within just 15 min after injection, followed by a return to baseline by 30 min. This rapid rise and subsequent decline suggest swift NMN metabolism. Notably, NAD+ levels continued to rise steadily for 60 min, highlighting the ongoing utilization of NMN or its metabolites for NAD+ synthesis [50]. 

When administered intravenously (50 mg/kg), NMN bypasses enteric and hepatic metabolism, enabling a small portion of intact NMN to directly participate in NAD+ synthesis within the liver and kidneys. While incorporation of intact NMN was still limited, the observed differences highlight the crucial role of administration methods on NMN bioavailability [29].

### 3.4. Bloodstream Dynamics and Tissue Distribution

Accurately measuring NMN in the bloodstream remains a significant hurdle, mirroring the challenges faced with defining cellular transport of NMN. Limitations in current methods and the lack of a gold standard, leading to conflicting reports. Studies show NMN detection ranging from undetectable levels to 90 µM after oral intake [22,33,51], hindering our understanding of NMN’s fate within the bloodstream. In general, studies using HPLC detect higher levels of NMN [50], whereas those using LC-MS/MS report much lower or undetectable levels of NMN [22]. 

While NMN did not directly cross the blood-brain barrier in mice after IV administration, its impact on brain NAD+ levels has proven to be substantial [29]. NMN administration (500 mg/kg) via IP injection significantly increased hippocampal NAD+ levels by 34–39% within just 15 min [52], with similar effects observed in the hypothalamus [53]. Even a lower dose, NMN (62.5 mg/kg) sustained hippocampal mitochondrial NAD+ levels for 24 h [54]. Intravenous NMN also showed minimal muscle uptake [29]. However, a clinical trial providing oral NMN to prediabetic women revealed elevated levels of proteins associated with insulin sensitivity in muscle tissues. Although muscle NAD+ levels remained unchanged, elevated NAD+ metabolites suggested a potential increase in muscle NAD+ turnover [55]. A critical knowledge gap remains, however: how long do tissue NAD+ levels remain elevated after supplementation ceases in humans? Addressing this question is crucial for understanding the long-term effects of NMN.

The chosen administration route significantly impacts NMN’s fate. NMN (500 mg/kg) administered via IP injection in mice effectively boosted NAD+ levels in the liver, kidney, white adipose tissue, pancreas, and heart, with the liver showing the most prominent increase. Oral gavage (500 mg/kg) yielded similar effects in the liver but showed less efficacy in all other tissues [46]. Clinical trials show oral supplementation increases plasma NAD+ for at least 24 h, but levels return to baseline within a month of discontinuation [56]. In a pilot study, the administration of 300 mg IV NMN in adults not only safely elevated NAD+ levels but also uniquely reduced triglycerides. A triglyceride-lowering effect has not been observed with oral NMN, indicating route-specific metabolic variations [57,58].

## 4. Functional Diversity of NAD+ Precursors

Despite sharing interconnected pathways, NAD+ precursors display diverse fates and functions, suggesting they are not simply interchangeable building blocks for NAD+ synthesis. Ongoing discoveries reveal their complexity, with each finding sparking new questions. Nevertheless, significant differences in precursor metabolism and bioavailability have been identified, highlighting their potential for tailored therapeutic interventions.

### 4.1. Efficacy and Therapeutic Applications of NAD+ Boosting Strategies

Although NAM is the principal contributor to basal NAD+ levels in mammals [29], its therapeutic efficacy is limited. Unlike NMN and NR, NAM can inhibit sirtuins, a class of proteins linked to NAD+ benefits [59,60]. Additionally, NAM conversion to NAD+ has inherent limitations. The initial step requires NAMPT, an energy-dependent enzyme subject to feedback inhibition, limiting its ability to substantially elevate NAD+. Elevating dietary NAM also results in a corresponding increase in NAM methylation and excretion, thereby diminishing its NAD+-boosting efficacy [61].

Nicotinic acid (NA) stands apart from other NAD+ precursors due to its distinctive flushing effect, even at low doses of 50 mg [62]. This side effect arises from NA’s direct activation of the GPR109A receptor, which is independent of processes associated with NAD+ biosynthesis [63,64]. Furthermore, NA exhibits lower efficacy in elevating NAD+ levels compared to NMN, NR, and NAM, rendering NA less enticing as an NAD+ booster [37].

NMN and NR are widely acknowledged as promising NAD+ boosters, raising NAD+ blood levels by 1.5–2.5-fold in clinical trials [57,65,66]. However, metabolic tracing studies reveal that NMN and NR are mostly metabolized to NA or NAM in the intestine [39,40]. While common logic suggests they would have similar effects and limitations as NA or NAM, they exhibit unique advantages despite their conversion. Notably, they lack the flushing side effects associated with nicotinic acid (NA) and do not appear to inhibit sirtuins, as seen with NAM [30,47,67,68,69]. Additionally, comparative studies showed that oral NR administration was more effective than NAM or NA at raising NAD+ levels in the liver of mice, and NMN was retained in the body longer than NAM [37,70]. Direct NMN transport to tissues could theoretically yield a more substantial increase in NAD+ production compared to NAM, as it bypasses the NAMPT-catalyzed reaction and is not subject to feedback inhibition.

Isotope labeling studies revealed an unexpected mechanism for NAD+ elevation. While NMN and NR are thought to directly elevate NAD+ as precursors, oral administration of labeled versions surprisingly led to an increase in unlabeled NAD+ metabolites [37,40]. This suggests indirect effects on endogenous NAD+ biosynthesis, potentially through activation of an unknown signaling pathway. Further investigation is needed to elucidate this novel mechanism and its contribution to the unique effects of NMN and NR. NMN offers a potential advantage over NR due to its direct cellular uptake, independent of NRK1/2 enzymes. This tissue-specific requirement for NRK1/2 conversion could limit NR’s efficacy in certain cell types, highlighting the potential therapeutic benefit of NMN in scenarios where NR might be less effective [22,48,49].

More recently, researchers have explored the effects of enzymes related to NAD+ synthesis, namely ACMSD, as possible therapeutic targets for boosting NAD+ levels. The enzyme α-amino-β-carboxymuconate-ε-semialdehyde decarboxylase (ACMSD) governs the rate at which ACMS is cyclized to form QA and continue the *de novo* synthesis cycle [71]. In both *C. elegans* and murine models, the inhibition of ACMSD was shown to increase NAD+ synthesis in a dose-dependent manner and subsequently increase SIRT1 and improve mitochondrial function [72]. This mechanism appears to be conserved evolutionarily between these models suggesting that the same may apply to humans, but further research is necessary to determine the extent of this link.

### 4.2. Physiological Context and Precursor Efficacy

The therapeutic efficacy of different precursors can diverge even within the same physiological context. For example, NR and NMN effectively promote hematopoiesis, but NA and NAM do not [73,74]. Oral supplementation with NR, but not NAM, restored NAD+ levels in the heart and improved cardiac function in a mouse model of cardiomyopathy [75]. While NMN supplementation has been shown to improve cardiac function in a model of Friedreich’s ataxia cardiomyopathy [76], similar effects were not observed for NR [77].

Aging disrupts NAD+ balance, prompting the intestine to upregulate the NMN transporter Slc12a8. This regulatory mechanism forms a feedback loop, enabling NMN supplementation to effectively restore NAD+ levels [42]. Conversely, with age and disease, there can be a decline in NAMPT activity and expression, which plays a crucial role in the salvage pathway for NAD+ production by converting NAM to NMN [67,78,79]. This downstream bottleneck impedes the efficacy of NAM supplementation for NAD+ repletion under such conditions. In contrast, NMN bypasses the NAMPT step in the salvage pathway, demonstrating superior capability to elevate NAD+ levels and alleviate dysfunction associated with NAMPT deficiency [67,80,81]. 

NRK2, a key enzyme in the salvage pathway for NAD+ biosynthesis, exhibits stress-induced upregulation. This is observed in injured neurons, muscle subjected to injury or high-fat diets, and stressed cardiac tissue [82,83,84,85]. This suggests a protective mechanism, as NR or NMN supplementation effectively reduces disease severity in most conditions that exhibit stress-induced elevations in NRK2 [82,86,87].

## 5. Human Studies Discussion

Human trials exploring NMN supplementation have yielded promising results. However, the outcomes exhibit notable inconsistencies, underscoring areas for improvement and guiding future research endeavors towards resolving these discrepancies. Understanding the nuanced factors influencing NMN’s efficacy—such as timing, dosage, and target populations—is essential to characterize its clinical significance. 

### 5.1. Modulation of NAD+ Levels 

Human trials measuring NMN’s impact on serum NAD+ levels show mixed results, highlighting significant challenges in measuring this metabolite (Table 1). While Yi et al. reported consistent dose-dependent increases in serum NAD+ and NMN, Katayoshi et al. found undetectable levels [88,89]. Further complicating the picture, Huang et al. observed a non-significant increase in serum NAD+/NADH, contradicting prior studies demonstrating efficacy at similar or lower doses [90]. 

Investigations into the impact of NMN supplementation on NAD+ concentrations reveal distinct NAD+ metabolic profiles within various blood cell fractions. Yamane et al. observed a significant increase in plasma NAD+ levels following daily NMN supplementation (250 mg) for 3 months in healthy volunteers, peaking within the first month and remaining elevated throughout supplementation. Similarly, Okabe et al. noted a doubling of NAD+ levels after one month of NMN supplementation, with sustained elevation until supplementation cessation. Notably, Okabe et al. assessed NAD+ in whole blood, capturing both circulating and cellular NAD+, while Yamane et al. focused on plasma, crucial for systemic distribution [56,91]. This discrepancy elucidates markedly lower NAD+ concentrations in plasma reported by Yamane et al., approximately 100 times lower than in whole blood. Intriguingly, NMN concentration was tenfold higher when measured in isolated plasma than when measured in whole blood by Okabe et al., highlighting the importance of sample type in NAD+ metabolism data interpretation. 

Supplementation appears to influence NAD+ metabolism beyond simple conversion to NAD+. Okabe et al. observed increases in NAMN, a deamidated metabolite of NMN, alongside elevated NAD+ [56]. Igarashi et al. also reported increased levels of NAD+ precursors (NR, NAR) after NMN supplementation, mirroring findings in animal studies [66].

NAD+ levels are dynamic, and influenced by diet, activity, and circadian rhythms. Studies suggest short-term fluctuations and a return to baseline within 2 h after daytime NMN administration. Long-term monitoring indicates stable NAD+ levels except for NMN-induced increases and exercise-related elevations [92]. Intravenous NMN transiently elevates NAD+ and activates SIRT1/CD38 pathways [58]. 

Interestingly, NMN’s effects on NAD+ levels appear tissue specific. Studies by Yoshino et al. and Pencina et al. observed increased NAD+ in PBMCs (blood cells) but not muscle tissue following NMN supplementation [55,93]. Furthermore, Qiu et al. showed reduced PBMC NAD+ in hypertensive patients, with NMN supplementation partially restoring these levels alongside lifestyle changes [94]. These findings highlight the complexity of NMN’s influence on NAD+ metabolism and the need for further research on its tissue-specific effects and potential interactions with other interventions.

### 5.2. Sleep Regulation and Quality

Studies evaluating the effects of NMN on sleep using single doses (100–500 mg) and short-term supplementation (250 mg for 8 weeks) showed no significant changes in sleep measures [57,95]. However, a longer-term study (12 weeks) showed hints of potential benefit. Afternoon NMN intake (250 mg) in this study reduced drowsiness in older adults, suggesting timing of the dose may play a role [40]. In a population with pre-existing sleep difficulties, 12 weeks of daily NMN supplementation (180 mg) yielded statistically significant improvements in sleep quality, latency, and daytime function [96]. These findings suggest NMN’s effect on sleep may depend on dose, duration, and potentially time of administration, with greater benefit observed in those with existing sleep issues.

### 5.3. Physical Performance

While some studies report improved gait speed, grip strength, and lower limb function [66,97], others show no significant effects on walking distance or in diabetic/obese populations [90,93,98]. Interestingly, NMN can improve muscle insulin sensitivity without impacting mitochondrial function or overall performance [55], suggesting a more nuanced effect on muscle health. Additionally, higher NMN doses (600–1200 mg) might benefit trained athletes in terms of aerobic capacity [99]. Timing of administration (afternoon vs morning) may also influence outcomes [97]. Overall, the evidence for NMN’s efficacy in enhancing physical performance remains inconclusive and warrants further investigation.

### 5.4. Cardiometabolic Health 

Human trials investigating NMN’s impact on cardiovascular health parameters yield mixed results. Several studies observed encouraging results, including reductions in weight, blood pressure, and cholesterol levels [89,93,94]. However, others have reported no significant changes in insulin sensitivity, lipid profiles, or vascular function [66,90]. Notably, positive effects on blood pressure and arterial stiffness were observed in hypertensive individuals and those with higher baseline glucose/BMI [89,94], suggesting potential benefits in specific subpopulations. It is important to note that larger and longer-term clinical trials with more defined populations are needed to make comparisons to established therapeutic approaches to cardiometabolic health are essential to evaluate the relative clinical value of NMN and guide future research efforts. Larger, more comprehensive studies with diverse participant groups will allow for better comparisons with existing treatments for heart and metabolic health. These studies are essential to evaluate the relative clinical value of NMN and guide future research efforts.

### 5.5. Glucose Metabolism and Regulation

In prediabetic women, 250 mg/day NMN significantly enhanced muscle insulin sensitivity, but these effects were not observed systemically [55]. Additionally, a small trial in postmenopausal women receiving 300 mg/day NMN for 8 weeks reported reduced HbA1c and increased adiponectin, suggesting potential anti-inflammatory and insulin-sensitizing properties [100]. Supplementation of NMN in healthy adults led to a significant increase in postprandial insulin levels after two months, suggesting enhanced glucose-stimulated insulin secretion [67]. However, the significance of this finding in healthy individuals remains uncertain [101]. Adding another layer of complexity, a large cross-sectional study involving over 1394 adults (most with existing metabolic conditions) found that those with the highest NAD+ levels had a three times greater risk of metabolic syndrome compared to those with the lowest levels [102]. These contradictory findings highlight the need for further research, particularly long-term studies, to understand NMN’s sustained effects on various health markers. Additionally, the studies involved diverse populations with different metabolic conditions. The observational study [102] suggests a potential link between NAD+ levels and specific metabolic profiles, emphasizing the importance of individualized approaches.

### 5.6. Overall Well-Being and Quality of Life

A clinical trial showed significant improvements in subjective general health scores at day 60 for all NMN doses, with earlier improvements (day 30) in higher dose groups [88]. While another study did not reach statistical significance, it observed a trend towards improvement in the NMN group [103]. Postmenopausal women in a small NMN supplementation study (8 weeks) reported subjective improvements in allergies, joint pain, overall well-being, recovery, cognitive clarity, and hair quality [100].

### 5.7. Telomere Lengthening

A small-scale study demonstrated significant telomere lengthening in PBMCs of male volunteers after 30 days of NMN (300 mg/day) supplementation. Telomeres continued to elongate at 60 days and nearly doubled from baseline by 90 days of supplementation [104]. NMN’s impact on telomeres may be linked to stabilizing telomeres and preventing tissue damage through its effect on NAD+ and the SIRT-1 pathway [105].

### 5.8. Side Effects and Safety Considerations

A growing body of human trials, currently encompassing 19 published studies, has investigated the safety of NMN supplementation. Early studies focused on single doses, with Irie et al.’s work demonstrating no adverse effects following ingestion of up to 500 mg NMN [57]. Subsequent research has explored chronic administration, with studies reporting good tolerability for doses ≤500 mg over at least a month. Higher doses (600–1250 mg) administered for up to 6 weeks have also shown no significant side effects in Liao et al. and Fukamizu et al.s’ trials [69,99]. Notably, Pencina et al. observed positive health outcomes alongside increased NAD+ levels in a study using 1000 mg NMN twice daily for 2 weeks [93]. 

While existing evidence is encouraging, a definitive understanding of NMN’s safety profile is still under development. Limitations in current research include short durations, small sample sizes, and a lack of diverse participants. Additionally, the potential for long-term effects, interactions with medications, and safety in individuals with pre-existing conditions require further exploration. To definitively establish a Tolerable Upper Intake Level (UL), long-term studies with age-specific considerations are crucial. Overall, NMN appears safe at moderate doses in healthy individuals, but ongoing, long-term research is essential to ensure its safe application in a wider population. Animal studies offer a glimpse into the potential hazard of increasing NMN as more comprehensive trials have been conducted using mice. It is reported that there is no carcinogenicity [67,106] or tumorigenicity [33,107] associated with prolonged NMN exposure for the purposes of increasing NAD levels. It seems, however, that during cancer progression, NAD boosting has shown deleterious effects by promoting cell survival, growth and propagation resulting in resistance to typical treatments and increased levels of inflammation [108,109]. This possible discrepancy between humans and mouse models can only be remedied with comprehensive long-term studies.

**Table 1 metabolites-14-00341-t001:** Clinical Trials Investigating NMN Supplementation.

Group Treated	NMN Dose	Duration	Sample Type	NAD+ Metabolome Levels	Functional Outcomes	Reference
Healthy Middle-Aged Japanese Men (40–60)	Placebo *n* = 13	8 Weeks	PMBCs (pmol/mg) HPLC	NAD: 26.83	↑ NAD+ in PBMCs Modest ↓ postprandial hyperinsulinemia	[95]
	250 mg daily			NAD: 41.43		
Healthy Adults (40–59)	Placebo 2× Daily *n* = 18	12 Weeks	Blood Serum (ng/mL) HPLC Tandem MS	NAM: 10.9 ± 4.8 NMN: <2 NAD: <5	↑ NAM in PBMCs ↓ CVD risk	[89]
	125 mg NMN 2× Daily *n* = 18			NAM: 16.5 ± 6.3 NMN: <2 NAD: <5		
Overweight Adults (45+)	Placebo 2× Daily *n* = 9	28 Days	Whole Blood (ng/mL) HPLC Tandem MS	NAM: 21.2 ± 7.77 NMN: 0.6 ± 0.08 NAD: 17.8 ± 2.77 1-MeNAM: 18.7 ± 7.42 2-PY: 394.7 ± 248	↑ Circulating NAD+ ↓ Weight, Cholesterol, BP	[93]
	1 g NMN 2× Daily *n* = 21			NAM: 19.7 ± 9.73 NMN: 0.7 ± 0.11 NAD: 19.4 ± 2.62 1-MeNAM: 17.9 ± 7.2 2-PY: 357.8 ± 151		
Patients Diagnosed with Mild Essential Hypertension (18–80)	LM Group *n* = 10	6 Weeks	PMBCs (pmol/mg) HPLC-MS	NMN: ~4.5 NAD: ~15	↑ NAD+ PBMCs (43%) ↑ ATP	[94]
	800 mg NMN + LM *n* = 9			NMN: ~6 NAD: ~20		
Healthy Adults (20–80)	Placebo *n* = 25	30 Days	Whole Blood (µM) Custom BRET NAD Sensor Assay and HPLC-MS	NAD: 23.8 ± 5.5	↑ Blood NAD+ Dose-dependent ↑ 2/4-PY (esp. 1000 mg NMN)	[92]
	500 mg NMN Daily *n* = 25			NAD: 41.7 ± 13.0		
	1000 mg NMN Daily *n* = 25			NAD: 58.8 ± 21.1		
Healthy Adults (20–80)	Placebo + Exercise *n* = 21	30 Days	Whole Blood (µM) Custom BRET NAD Sensor Assay and HPLC-MS	NAD: 33.18 ± 7.2	Exercise & NMN: ↑ NAD+ PBMCs (similar to 1000 mg NMN)	[92]
	500 mg NMN + Exercise *n* = 21			NAD: 55.48 ± 21.4		
Healthy Adults (20–65)	250 mg NMN Daily *n* = 11	12 Weeks	Blood Plasma (µM) HPLC-MS	NMN Month 1: ~0.15 Month 4: ~0.5 NAD Month 1: ~0.055 Month 4: ~0.01	↑ NMN & NAD in plasma Transient ↑ insulin (2 mo) NMN ↑ (3 mo) NAD peak (1 mo)	[91]
Males with Diabetes and Reduced Grip Strength of Walking Speed (65+)	Placebo *n* = 8	24 Weeks	-	Not Measured	NMN: ↓ Frailty	[98]
	250 mg NMN Daily *n* = 8					
Healthy Adults (20–65)	Placebo *n* = 15	4 Weeks	-	Not Measured	Safe & Well-Tolerated	[69]
	1250 mg NMN Daily *n* = 16					
Healthy Adults (40–65) BMI (18.5–35 kg/m^2^)	Placebo *n* = 31	60 Days	Serum NAD/NADH (pmol/mL) Colorimetric Quantitation Kit	NAD: 8.14 ± 4.86	↑ Serum NAD (11.3%) ↓ HOMA-IR ↑ Walking Endurance Improved Well-being	[90]
	300 mg NMN Daily *n* = 31			NAD: 9.07 ± 5.65		
Healthy Men (65+)	Placebo *n* = 21	12 Weeks	Whole Blood (µM) LC-Tandem MS	NAM: 10.6 ± 1.6 NMN: 0.105 ± 0.013 NAD: 0.53 ± 0.12 NAMN: 0.05 ± 0.03 NR: 0.0308 ± 0.0088 NA: 0.00684 ± 0.00168	↑ NMN, NAD+, NR, NAMN Improved Gait, Walk, Grip (nominal)	[66]
	250 mg NMN Daily *n* = 21			NAM: 13.4 ± 2.4 NMN: 0.127 ± 0.019 NAD: 1.07 ± 0.16 NAMN: 3.51 ± 1.86 NR: 0.0549 ± 0.0241 NA: 0.00974 ± 0.00129		
Adults (65+)	Placebo-AM *n* = 27	12 Weeks	-	Not Measured	Afternoon NMN: ↑ Limb Function ↓ Drowsiness (Older Adults)	[97]
	Placebo-PM *n* = 27					
	250 mg NMN-AM *n* = 27					
	250 mg NMN-PM *n* = 27					
Healthy Individuals (20–70)	Intravenous administration 300 mg NMN in saline (3 mg/mL) Daily *n* = 10	5 h	Whole Blood NAD/NADH Assay Kit	1.2-fold increase in Total NAD	IV NMN: Safe, ↑ Blood NAD+ ↓ Triglycerides	[58]
Post-Menopausal Women (50–80)	300 mg NMN Daily *n* = 16	8 Weeks	Whole Blood (ng/mL) HPLC-MS	NAM: 164.7 ± 20 NMN: 1.28 ± 0.2 NAD: 13.7 ± 2	NAM ↑; NAD ↓; NMN ↔	[100]
Healthy Adults (22–64)	Placebo *n* = 15	12 Weeks	Whole Blood (µM) HPLC-MS	NAM: 19 ± 3 NMN: 0.055 ± 0.01 NAD: 22 ± 2 NAMN (8 Wks): 0 NR: 0.04 ± 0.025 NA: 0.25 ± 0.06	Blood NAD+ ↑, NAMN ↑, (NMN, NA, NAR, NAAD, MNAM) ↔	[56]
	125 mg NMN 2× Daily *n* = 15			NAM: 17 ± 8 NMN: 0.054 ± 0.016 NAD: 45 ± 20 NAMN: 2.0 ± 1.5 NR: 0.045 ± 0.005 NA: 0.26 ± 0.04		
Overweight Adults (55–80)	Placebo 2× Daily *n* = 8	14 Days	Whole Blood (ng or µg/mL) HPLC Tandem-MS	NMN: 0.0326 (µg) NAD: 1.36 (µg) NAM: 10.2 1-MeNAM: 7.57 2-PY: 103 NR: 0.406	↑ Blood NAD Metabolites ↓ Body Weight, Systolic BP, ↓ Diastolic BP	[110]
	1000 mg NMN 1× Daily *n* = 12			NMN: 0.0882 (µg) NAD: 23.0 (µg) NAM: 65.2 1-MeNAM: 146 2-PY: 2150 NR: 1.30		
	1000 mg NMN 2× Daily *n* = 12			NMN: 0.148 (µg) NAD: 40.4 (µg) NAM: 140 1-MeNAM: 276 2-PY: 4230 NR: 1.48		
Healthy Adults (37–50)	Placebo *n* = 20	60 Days	Blood Serum (nM) Colorimetric Quantitation Kit	Day 0: 8.11 ± 5.16 Day 30: 9.83 ± 8.43 Day 60: 11.8 ± 9.4	Blood NAD: ↑ (all NMN groups, days 30 & 60)	[88]
	300 mg NMN *n* = 20			Day 0: 11.8 ± 11.7 Day 30: 29.8 ± 20.1 Day 60: 32.6 ±17.9		
	600 mg NMN *n* = 20			Day 0: 7.95 ± 3.29 Day 30: 39.0 ±12.6 Day 60: 45.3 ±11.8		
	900 mg NMN *n* = 20			Day 0: 10.5 ± 6.8 Day 30: 43.1 ± 14.3 Day 60: 48.5 ±19.8		
Young and Middle-Aged Recreational Runners	Placebo *n* = 12	6 Weeks	-	Not Measured	Exercise Capacity ↑ (likely ↑ O_2_ utilization in skeletal muscle)	[99]
	300 mg NMN *n* = 12					
	600 mg NMN *n* = 12					
	1200 mg NMN *n* = 12					
Overweight Post-Menopausal Women BMI (25.3–39.1 kg/m^2^)	Placebo *n* = 12	10 Weeks	PMBCs (pmol/mg)	NAD: ~25	↑ 2-PY & 4-PY; ↑ PBMC NAD+ (43% vs placebo); ↑ Muscle NAD+ turnover & ↑ Muscle Insulin Sensitivity (25%)	[55]
	250 mg NMN *n* = 13			NAD: ~40		
Healthy Men (40–60)	100 mg NMN *n* = 10	12 Weeks	Blood Plasma (nM) HPLC Tandem MS	2-PY: ~2000 4-PY: ~350 1-MeNAM: ~225	↑ Bilirubin (51.3%) ↓ Glucose (11.7%), Creatinine (5.1%), Chloride (2.3%)	[57]
	250 mg NMN *n* = 10			2-PY: ~2500 4-PY: ~400 1-MeNAM: ~250		
	500 mg NMN *n* = 10			2-PY: ~4000 4-PY: ~750 1-MeNAM: ~300		

## 6. Remaining Questions and Future Directions

As the field of NMN research continues to evolve, several key questions remain unanswered, pointing toward exciting future avenues of investigation. In this section, remaining uncertainties surrounding NMN and potential directions for future research are outlined, offering insights into the promising perspectives that lie ahead.

### 6.1. Sex-Specific Differences in Response to Supplementation

Women tend to have lower whole blood NAD+ levels compared to men, as observed in clinical trials [92,111], suggesting sex-specific differences in NAD+ metabolism and its role in health. A study evaluating the influence of hormones on NAD+ levels revealed that a decrease in testosterone was strongly correlated with a downregulation of NAD+ production [112]. Preclinical models on NMN further support the significance of sex-related differences. In BESTO mice engineered for increased β-cell Sirt1, NMN supplementation restored the age-impaired protective effect of Sirt1 on insulin secretion and glucose tolerance only in females [113]. Similarly, NMN treatment reversed the significant decline in NAD+ levels and impaired glucose tolerance observed in female, but not male, NAMPT-heterozygous mice [81]. Interestingly, sex divergence extends beyond age-related changes. Even in baseline and stroke conditions, NAD+ levels differ between male and female mice [114]. This sex-specificity is further highlighted by the lifespan extension observed upon NAMPT overexpression, which was exclusive to female mice [115].

### 6.2. Individual Variability in Supplementation Effects

A recent study revealed significant heterogeneity in individual responses to NMN supplementation that may be attributed to underlying differences in gene expression patterns [92]. Individuals who exhibited a robust increase in NAD+ levels upon NMN supplementation (“responders”) demonstrated higher expression of NAD+-synthesizing enzymes. Conversely, “non-responders” displayed a stronger expression of NAD+-consuming enzymes. This variability may explain some inconsistencies observed in human trials. This finding underscores the potential of a therapeutic approach that targets both NAD+ synthesis and degradation to achieve more robust and consistent elevations in NAD+ levels, especially for “non-responders.” Lifestyle interventions such as exercise and calorie restriction may offer benefits for individuals with high NAD+ consumption, potentially by rebalancing NAD+ homeostasis [92,116]. Trials targeting multiple aspects of NAD+ metabolism have shown benefits, but still exhibit individual variability in response [117]. A combined approach is likely optimal. Boosting NAD+ synthesis while controlling its breakdown through various methods might be the most effective way to increase NAD+ levels, particularly for individuals with high NAD+ consumption.

### 6.3. Precursor-Independent Regulatory Functions

Beyond its established role as an NAD+ precursor, NMN demonstrates emerging regulatory functions in mitochondrial [54,118,119] and neuronal [120,121,122] metabolic processes. Numerous preclinical studies demonstrate NMN’s neuroprotective potential, enhancing cognition, reducing synaptic loss, improving brain mitochondrial function, and lowering inflammation [123,124,125,126,127,128,129]. However, a 2015 study raised concerns about NMN accumulation potentially causing axonal degeneration [130]. This effect appears to be mediated by SARM1, a NADase activated by NMN under low NMANT2 conditions (stress/injury) [79,131,132]. The key factor seems to be the NMN/NAD+ ratio, not absolute NMN levels [131]. NMN-mediated neuroprotection coincides with increased NAD+, suggesting a crucial balance [133,134]. While NMNAT2 levels may decrease with age (potentially increasing vulnerability), recent clinical trials suggest NAM, another NAD+ precursor, is safe and effective for glaucoma, a neurodegenerative disease [135,136]. Further research is needed on NMNAT2 levels, NMN/NR vulnerability, and long-term clinical trials.

NMN may influence cellular health by impacting mitochondrial DNA (mtDNA) replication. Studies suggest NMN sustains mitochondrial nucleotide levels, leading to increased mtDNA replication and NAD+ metabolite levels [137]. Additionally, HINT2, a protein with an NMN binding site, may improve mitochondrial NAD+ levels and cell survival [138,139]. However, research is limited to specific cell types, and the impact on enzymatic activity needs further exploration.

These models exemplify the need to carefully consider the metabolic roles of NAD+ precursors when designing and implementing NAD+ boosting strategies—particularly those that increase the permeability of the precursors [140,141] to mitigate undesirable and unintended effects as NAD+ metabolism isn’t as straightforward as once thought.

### 6.4. Influence of Gut Microbiota

Significant knowledge gaps remain regarding the intricate interplay between gut microbiota and NAD+ precursor metabolism. For example, a study in mice showed that dietary fiber supports bacterial NAD+ synthesis, but the impact of different fiber types and gut health on NAD+ synthesis pathways is not clear [142]. This study also uncovered a reciprocal exchange of precursors vital for NAD+ production, with potentially distinct metabolic outcomes in different segments. This regional variation is often overlooked in metabolic analyses. Multiple studies highlight the puzzling rise of unlabeled metabolites after labeled NMN supplementation, indicating an indirect effect on endogenous biosynthesis pathways. Potential explanations have been proposed, including altered enzyme kinetics, conversion of NMN to NR “reservoirs”, or the reversal of the NMNAT reaction. While the mechanism has not been determined, the distinct metabolite profiles of germ-free vs. colonized mice strongly suggest gut bacteria play a key role in this process. Notably, the distinct metabolite profiles of germ-free vs. colonized mice strongly implicate gut bacteria in this process [40]. Tracing labeled metabolites in mouse fecal matter could shed light on the dynamics of this competition and elucidate the role of NMN in the microbiome. Furthermore, identifying signaling pathways affected by antibiotic treatment in mice could clarify the mechanisms underlying the observed differences in precursor elevations through amidated and deamidated pathways.

### 6.5. Optimizing Bioavailability

Oral drug delivery revolutionized healthcare by offering a convenient and well-tolerated route for treatment [143]. However, challenges such as instability, poor water solubility, and physiological barriers hinder the efficacy of some compounds. Gastric acid, proteolytic enzymes, and first-pass metabolism in the liver further reduce bioavailability [144,145]. Similar challenges exist with other administration routes like intramuscular and intravenous injections, leading to significant interest in alternative strategies to improve bioavailability [146].

Emerging research suggests that reduced forms of NAD+ precursors show superior stability and enhanced efficacy in elevating NAD+ levels compared to their oxidized counterparts. Notably, nicotinamide riboside hydride (NRH) demonstrates potency exceeding both NR and NMN, increasing NAD+ levels up to 10-fold in various cell lines [118] NRH utilizes distinct biochemical pathways from NMN and NR, entering cells via ENTs and converting to NMNH independently of NAMPT and NMNAT [147,148]. NMNH exhibits potent NAD+ boosting, surpassing NMN in speed and magnitude, and offers additional benefits like promoting kidney cell healing after injury. Increased NAD+ levels were detected in key organs including the kidney, liver, muscle, brain, brown adipose tissue, and heart [149]. However, these studies underscore the need for a more comprehensive understanding of extreme elevations in NAD+ levels. For instance, supplementation with NRH in HepG3 cells, but not HEK293 cells, resulted in cytotoxicity, likely induced by oxidative stress [150]. Elucidating the mechanisms underlying this toxicity and the reasons for cell-type specificity is crucial for ensuring the safety of NAD+ precursor interventions.

Liposomal delivery systems are commonly used to enhance the bioavailability of NAD+ precursors. These systems consist of microscopic vesicles composed of phospholipid bilayers, mimicking the structure of cell membranes [151]. The therapeutic agent is encapsulated and protected from degradation by enzymes and harsh conditions within the gastrointestinal tract. With structural similarities to cells lining the small intestine, liposomes can fuse with M cells within the small intestine, allowing entry into the lymphatic system and bypassing hepatic processing altogether [151]. Notably, studies in rats have demonstrated that the phospholipid bilayer of liposomes does not adversely affect blood lipid levels [152]. Encapsulated NMN is effectively taken up by cells in vitro and demonstrates enhanced absorption and tissue concentration in mice [153,154]. In a model of aging mice, oral administration of NMN encapsulated in ovalbumin-fucoidan nanoparticles significantly reduced oxidative stress, improved behavioral performance, and led to a 1.34-fold increase in NAD+ levels compared to free NMN administration [155]. The metabolic processing of common administration routes for NMN are depicted in Figure 3.

Our understanding of the complex metabolic pathways for NAD+ precursors is still developing. This makes it difficult to predict if bypassing or altering specific steps to enhance bioavailability might have unforeseen effects. For instance, NMN supplementation in mice lacking gut bacteria disproportionately increase plasma NR levels [40] The underlying mechanisms and downstream consequences of this increase are currently unknown. Similarly, while intravenous administration offers a potential route to bypass first-pass metabolism and enhance NAD+ precursor bioavailability, current data on this approach is limited to short-term studies [58]. Further knowledge on the significance of each step of the NAD+ metabolic pathway is crucial for tailoring bioavailability strategies for optimal and safe NAD+ levels.

### 6.6. Refining Strategies for Clinical Applications

Traditionally, research on NAD+ precursors have heavily relied on measuring blood NAD+ levels as a marker of efficacy due to its ease and practicality. However, it is crucial to acknowledge its limitations. Blood NAD+ levels do not necessarily capture the full picture of a precursor’s biological effects [76,77]. Studies show that even when NR and NMN achieve similar increases in blood NAD+, their impact on different tissues and organs can diverge [22,48,49]. Furthermore, the optimal dose for a specific benefit may not simply correlate with the highest NAD+ increase. For instance, various NR doses showed similar NAD+ elevation, but a lower dose was more effective in improving metabolic flexibility in obese mice [156].

Establishing definitive levels for low, optimal, and high NAD+ is crucial for designing effective interventions and interpreting study outcomes. The absence of a universally accepted definition for “optimal” NAD+ levels hinders our ability to draw clear conclusions from individual and comparative studies. This challenge is exacerbated by significant variations in baseline whole blood NAD+ levels, influenced by factors such as age, ethnicity, gender, and lifestyle [56,110,157]. Consequently, supplementation may result in vastly different effects in NAD+-deficient individuals versus those with relatively high baseline levels. This variability in response makes it difficult to draw definitive conclusions on the effects of NAD+ precursor supplementation in clinical trials. For instance, a post-hoc analysis of an NMN trial conducted by Yi et al. in 2023 [88] found a correlation between changes in NAD+ levels and improved walking test results, with an approximate 15 nmol/L increase in NAD+ [158]. The low-dose group in this trial did not show overall improvements in the walking test, possibly due to large variations in starting NAD+ levels, which likely obscured any potential benefits from the lower dosage. By incorporating a dose-response analysis, the median effective dose (ED50) required to achieve the desired outcome in 50% of the population can be determined. This approach represents a significant advancement over previous studies that focused solely on boosting NAD+ levels, as it enables the establishment of targeted dosage regimens tailored to specific therapeutic goals.

Current research highlights the importance of not just considering the effects of NMN and NR on various tissues, but also differences within cellular compartments. Distinct NAD+ pools exist in the cytosol, mitochondria, and nucleus, each serving specific functions [19]. The identification of separate transporters for NAD+ (SLC25A51) and NMN (Slc25a45) in mitochondria suggests a fascinating level of compartmentalized regulation within cells [43,159]. This implies that cells can precisely control the influx of these molecules based on specific needs, potentially offering a new avenue for understanding and influencing cellular health.

Sensitivity, specificity, and interference issues plague current approaches, impeding the interpretation of interventions and metabolic pathways. To address this, meticulous extraction protocols, stringent internal controls, and optimization of existing methods are crucial. Addressing these limitations while also investigating cutting-edge technologies ensures a path toward precise NMN detection.

## 7. Conclusions

NMN holds promise for modulating NAD+ levels, but its effectiveness in humans is limited by an incomplete understanding of its metabolism. Recent evidence suggests a complex interplay between NMN, gut microbiota, and individual metabolic profiles, contributing to inconsistent clinical trial findings. The increasing interest in NMN-mediated NAD+ modulation underscores the need for high-quality clinical trials with clearly defined NAD+ level thresholds and standardized analytical methods that consider patient-specific metabolic variations. Through collaborative research efforts that prioritize these areas, future NMN research can move towards a more comprehensive understanding of its efficacy and safety in humans.

## Figures and Tables

**Figure 1 metabolites-14-00341-f001:**
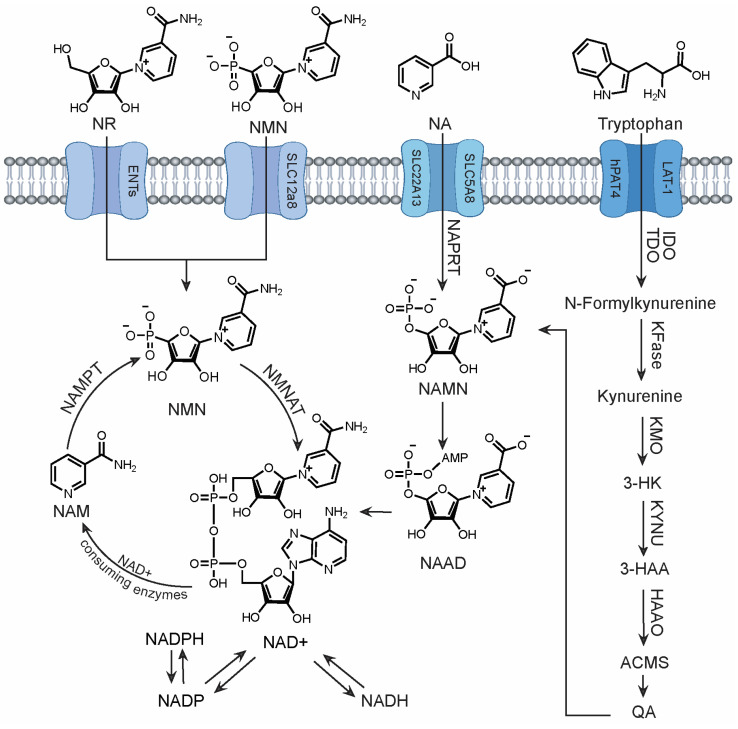
Biosynthetic pathways of NAD+ synthesis in mammalian cells. The salvage pathway is the body’s main and most efficient source of NAD+. Abbreviations: 3-HK, 3-hydroxykynurenine; 3-HAA, 3-hydroxy anthranilic acid; ACMS, 2-amino-3-carboxymuconic semialdehyde; QPRT, quinolinate phosphoribosyltransferase; NA, nicotinic acid; NAPRT, nicotinic acid phosphoribosyltransferase; NAMN, nicotinic acid mononucleotide; NMNAT, nicotinamide mononucleotide adenylyltransferase; NAAD, nicotinic acid adenine dinucleotide; NADS, NAD+ synthetase; NR, nicotinamide riboside; NMN, nicotinamide mononucleotide; NRK, nicotinamide riboside kinase; NAD+/NADH, nicotinamide adenine dinucleotide; NAM, nicotinamide; NAMPT, nicotinamide phosphoribosyltransferase; 3-HAAO, 3-Hydroxyanthranilate 3,4-Dioxygenase; NADP/NADPH, Nicotinamide adenine dinucleotide phosphate; TDO2, Tryptophan 2,3-dioxygenase; KYNU, kynureninase KFase, kynurenine formidase, KMO, Kynurenine 3-Monooxygenase.

**Figure 2 metabolites-14-00341-f002:**
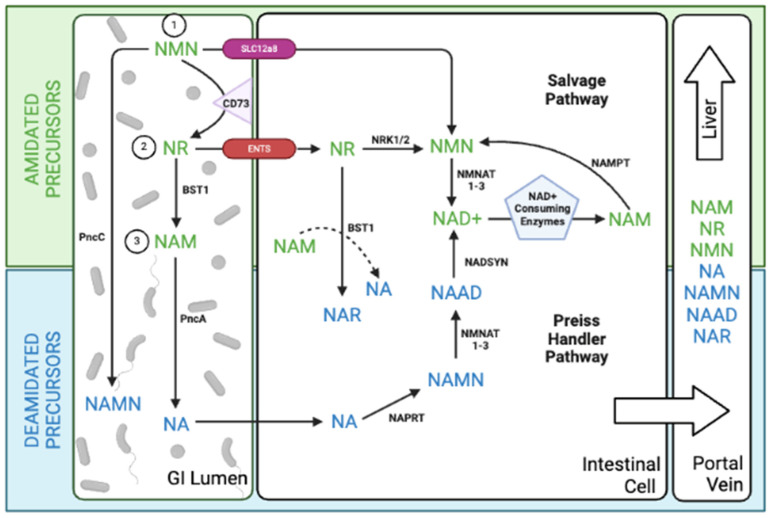
Metabolic fates of orally administered NAD+ Precursors. 1. NMN can be deaminated by the bacterial enzyme pncC to form NAMN. Alternatively, NMN may be broken down by other gut enzymes to produce NR or NAM, which is then metabolized by pncA to form NA and follow the Preiss Handler pathway. This deamidated route results in increased levels of NAR and NAMN in the gut and liver, and elevated NAAD specifically in the liver. A portion of oral NMN may follow the salvage route after direct entry through SLC12a8 or indirect entry through ENTs via conversion to NR. 2. During the early phase of metabolism, oral NR can directly enter cells and undergo the canonical salvage pathway. In the later phase, NR is converted into NAM by BST1, resulting in increased NA in the gut, elevated NA and NAR in the portal blood, and an increase in both amidated (NA, NAR, NAMN, and NAAD) and deamidated (NAM and NMN) metabolites in the liver. Additionally, NR may undergo a BST-1 catalyzed base-exchange reaction using NAM and NR, converting it to NAR. 3. NAM is rapidly converted to NA by the bacterial enzyme pncA in the colon, leading to elevation of deamidated precursors (NAMN, NAR, and NAAD) as well as amidated precursors (NAM and NMN) in the gut cell, portal vein, and liver. Figure created in BioRender.

**Figure 3 metabolites-14-00341-f003:**
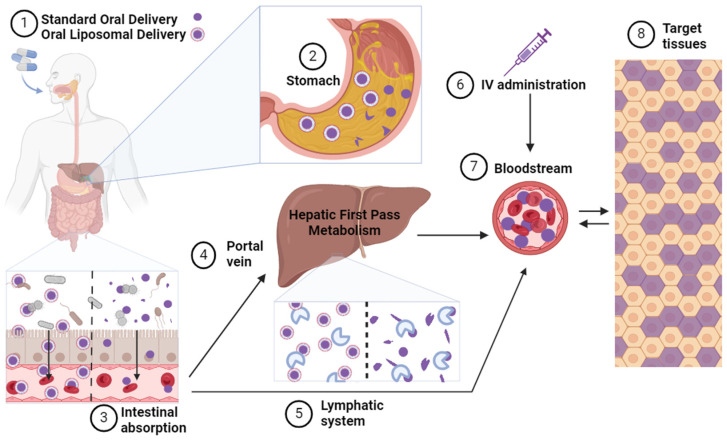
NMN Administration Routes and Metabolic Processing. (1) Following ingestion of either a standard or liposomal formulation of NMN, (2) the capsule undergoes breakdown in the stomach, with liposomal formulations offering enhanced protection from degradation. (3) The standard NMN formulation undergoes extensive metabolism in the intestine by gut bacteria and intestinal enzymes before absorption into the enterocyte, while liposomes fuse with the intestinal lining, facilitating the absorption of encapsulated NMN. From the enterocyte, (4) standard NMN is absorbed into the portal vein and transported to the liver, where it undergoes first-pass metabolism and further enzymatic breakdown. (5) absorbed liposomes exit the enterocytes via the lymphatic system and enter the bloodstream, bypassing first-pass hepatic metabolism. (6) Intravenous administration of NMN directly introduces NMN into the bloodstream without undergoing processing in the intestine or liver. (7) Ultimately, NMN molecules enter the bloodstream for potential uptake into target tissues, albeit the amount reaching these tissues varies depending on the route of administration and the metabolic processing experienced en route. Figure created in BioRender.

## Data Availability

Not applicable.

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
