# Peer review of "Nicotinamide Mononucleotide Supplementation: Understanding Metabolic Variability and Clinical Implications"

_metabolites, 2024, doi:10.3390/metabo14060341_

Round 1
Reviewer 1 Report
Comments and Suggestions for Authors
The present manuscript reviews NAD metabolism and NMN supplementation from the perspective of company scientists whose employer primarily sells NMN supplements. Overall, the paper is comprehensive and strikes a scholarly tone, although with the underlying assumption that NMN supplementation is good for health, which has not been proven for any indication or condition. The most important contributions of the paper are laying out the limited current clinical evidence in a thorough table (such tables may exist elsewhere, but I am not familiar with them) and relatively carefully covering the sometimes contradictory data regarding metabolism by microbiome, liver, and rest of body. The most concerning deficiencies are
1. Persistently referring to capturing or maximizing the benefits NMN, rather than evaluating whether such benefits exist; this is a pervasive problem that renders the manuscript in its current form unacceptable, with a particularly glaring example being “clinical trials evaluating the safety and efficacy of NMN in humans are underway and many echo the positive effects seen in animal models”
2. Taking a more critical approach to literature that argues against NMN being effectively absorbed intact and therefore having unique properties, than to literature showing the opposite
3. Not showing depth of chemical knowledge. For example referring to “dimeLC-MS/MS” as a distinctive method when it is use of isotopic internal standards, a common practice in the field of mass spec quantitation for NAD and many other metabolites; saying that NMN is lipophilic due to its phosphate group (NMN is not so lipophilic due to the phosphate group); saying that plasma NMN is 10x whole blood NMN, which is not possible as blood is 50% plasma (barring some additional reactions occurring, which may well be the case); describing controversy over the putative NMN transporter without evaluating the substance of the evidence and the likely truth (digging into these things--not job identifying them--is job of review writer)
4. Writing a large section on liposomes which is not merited and does not fit with the otherwise relatively scholarly content that is properly NAD/NMN focused
Minor issue: the large table has one line with NAD listed twice (at two different concentrations)--which is correct?
Author Response
Thank you for your time reviewing this manuscript and for providing thoughtful comments. Please find the detailed responses in the attached and the corresponding revisions/corrections in the re-submitted file.

Reviewer 2 Report
Comments and Suggestions for Authors
This manuscript provides a comprehensive review of the role of NMN in metabolic processes and its potential therapeutic implications. The discussions on the challenges of NMN bioavailability, its delivery strategies, and human clinical studies are particularly insightful and valuable. Overall, this is a well-constructed and informative review of the current knowledge of NMN that will interest a broad readership. I recommend this manuscript for publication in Metabolites with minor revisions as below.
In Figure 1, please add an arrow from QA to NAMN to clarify the pathway.
If the figures were created using Biorender.com, appropriate credit should be given, and the copyright should be cleared.
While the use of AI does not compromise the originality of the manuscript, the style and phrasing of many sentences suggest they may have been prepared with the aid of AI software. If so, should this be disclosed to adhere to the journal's publishing policies?
Author Response

(The authors gave the same response as above.)

Reviewer 3 Report
Comments and Suggestions for Authors
The manuscript “Nicotinamide Mononucleotide: Deciphering Metabolic Com-2 plexities for Improved Health Outcomes” by Candace Benjamin and Rebecca Crews is devoted to review the current knowledge on the effect of NAD+-supporting precursor supplementation on the health and active longevity of mammals. Besides, mechanisms of NAD+ synthesis, precursor delivery, and probable reasons for discrepancy in the reported data are also reviewed.
The main problem of this work is related to the fact that the emphasis is shifted from the main issues, such as the effect of NAD+ and its precursors on the longevity of rodents and other mammals, long-term risks of chronic administration, mechanisms of systemic effects of these compounds, mechanisms of stimulation of metabolism, etc, to secondary ones (drug delivery), which is obviously due to the honestly declared conflict of interest of Rebecca Crews, who is the Research Director at Renue by Science company, which produces various forms of NAD+ precursors.
The Review touches on, but does not address, the following important issues:
1. Does increasing NAD+ levels in the blood (tissue/blood cells/brain/muscle) of older animals actually help improve their physical condition and to what extent? How long does the improvement last, and is it associated with increased longevity? Have there really been long-term experiments assessing risks for older individuals?
2. What are the proposed mechanisms for the beneficial effects of NAD+ precursors? What are the effects observed in different systems?
3. If NAD+ precursors have a therapeutic effect in various pathologies, how does this effect compare in strength to the effects of standard drugs used to treat these pathologies?
4. What is the pharmacokinetics of NAD+ precursors with different routes of administration? How quickly do NAD+ levels decrease (especially in tissue) after supplementation is stopped?
Minor questions.
1. In the table, data on the level of NAD+ and precursors are given in different units (ng/mL, pmol/mL, μM, fold increase, pmol/mg, μg/mL, and nM). This makes it very difficult to compare the orders of magnitude obtained and masks their differences.
2. When precursors are administered orally, a significant portion of them is consumed by the microbiota. Will this cause the effectiveness of their application to decrease more and more? What is known about this issue?
3. The authors describe the benefits of delivering NMN into the blood via the lymph using liposomes manufactured by Renue by Science. Could the liposomes be the source of triglycerides in the blood? How much of triglycerides is introduced to blood with each portion of NMN? Why it is necessary to deliver NMN if one can effectively deliver NADH itself (Ref. 82)?
4. In Conclusions, authors write: “NMN possesses distinct functionalities, but its effectiveness in humans remains limited by an incomplete understanding of its metabolism.” May the effectiveness be limited by intrinsic mechanisms authors want to improve?
To conclude, еhe review could be interesting and useful for the reader, but the shift in emphasis, insufficient (or insufficiently structured) coverage of many important issues, and a rather superficial analysis of the data rather obscures than clarifies the overall picture of the results obtained to date.
Author Response

(The authors gave the same response as above.)

Round 2
Reviewer 1 Report
Comments and Suggestions for Authors
acceptable revision
Author Response
Thank you for taking the time to review our manuscript. We sincerely appreciate your thoughtful comments. Your feedback has been incredibly helpful in strengthening the scientific rigor of our work and ensuring we've clearly articulated our review.
Reviewer 3 Report
Comments and Suggestions for Authors
I carefully analyzed the Authors' answers to my questions and have to admit that in most cases they were too formal and meaningless to be real answers. I'm not satisfied with them.
But my primary concern is that the title, “Nicotinamide Mononucleotide: Deciphering Metabolic Complexities for Improved Health Outcomes,” and the abstract are actually misleading. Indeed, “further explorations are needed” or equivalents are the most common intermediate conclusions through the text for majority of paragraphs. This is obviously due to the controversy in the data obtained to date. However, the general message of the manuscript does not reflect this but suggests that the positive effects of various NAD+ precursors are firmly established and that the main question to be solved is the effective ways of precursor’s delivery.
Author Response
Thank you for taking the time to review our manuscript. We sincerely appreciate your thoughtful comments. Your feedback has been incredibly helpful in strengthening the scientific rigor of our work and ensuring we've clearly articulated our findings. Please find our response to your comments attached.
